# Commander Complex—A Multifaceted Operator in Intracellular Signaling and Cargo

**DOI:** 10.3390/cells10123447

**Published:** 2021-12-07

**Authors:** Saara Laulumaa, Markku Varjosalo

**Affiliations:** Molecular Systems Biology Group, Institute of Biotechnology, University of Helsinki, 00790 Helsinki, Finland; saara.laulumaa@helsinki.fi

**Keywords:** commander complex, COMMD, retromer, endosomal trafficking, endosome, cargo, WASH, NF-kB, cancer

## Abstract

Commander complex is a 16-protein complex that plays multiple roles in various intracellular events in endosomal cargo and in the regulation of cell homeostasis, cell cycle and immune response. It consists of COMMD1–10, CCDC22, CCDC93, DENND10, VPS26C, VPS29, and VPS35L. These proteins are expressed ubiquitously in the human body, and they have been linked to diseases including Wilson’s disease, atherosclerosis, and several types of cancer. In this review we describe the function of the commander complex in endosomal cargo and summarize the individual known roles of COMMD proteins in cell signaling and cancer. It becomes evident that commander complex might be a much more important player in intracellular regulation than we currently understand, and more systematic research on the role of commander complex is required.

## 1. Introduction

Cellular homeostasis maintenance in a trans-Golgi network is necessary for every aspect of cellular life including cell growth, cell cycle, cell death, various signal transduction pathways, and immune response [1,2,3]. One of the master regulators of this homeostasis is a recently discovered multiprotein Commander complex [4,5]. It regulates cell signaling via endosomal trafficking, transporting transmembrane channel proteins and receptors to the cell surface and back to the cytoplasm either for storage or to degradation in lysosomes.

Upon the discovery in a systematic protein-protein interaction (PPI) analysis by Wan et al. [2], its existence was confirmed shortly after in multiple contemporary interactome studies and phylogenetic profile analyses [6,7,8,9]. Commander complex consists of ten commander proteins (COMMD1–10), coiled-coil domain containing proteins CCDC22 and CCDC93, vacuolar protein sorting-associated proteins VPS26C, VPS35L, and VPS29 [5,10] and a DENN domain-containing protein 10 (DENND10) [8,11]. How these proteins assemble into a functional commander complex remains poorly understood.

Initially, the COMMD1 protein was linked to copper homeostasis regulation [12]. Later, COMMD1 were found to control Na+ homeostasis via regulating the human epithelial sodium channel (ENaC) [13], and the commander complex has been subsequently linked to the regulation of several ubiquitous fundamental cellular functions: ion and lipid homeostasis [14,15,16,17,18], embryogenesis [4], immune response [9,19,20,21,22,23], cell growth [24] and cell cycle [23,25,26,27]. Changes in COMMD protein function have been connected to developmental disorders [4] and several diseases like Wilson’s disease [28], Parkinson’s disease [29], atherosclerosis [17], viral protein recycling [10,30], and most importantly cancer [31,32,33].

In this review, we describe the function of commander complex in the endosomal cargo and summarize the known functions of individual COMMD proteins in the regulation of cell homeostasis regulating immune response, signal transduction and cell cycle. The role of COMMD1 is overrepresented in the past research it has been studied more extensively than other members in the complex [34]. It should be underlined that if a function has been reported for a certain COMMD protein, it is not necessarily specific for that individual protein but can be shared with other COMMD proteins. Many of these studies have focused on individual COMMD proteins, and therefore more systematic studies assessing the COMMD proteins a whole instead of individual entities are needed. Most of the reported roles of COMMD proteins are very recent individual discoveries, indicating that many important observations on this complex are still unmade. These will further underline the importance of the Commander complex and expand the already known diverse functions and roles of the complex.

## 2. Composition and Assembly of the Commander Complex

COMMD proteins 1–10 are sequence homologs named after COMMD1 [35]. COMMD1 was discovered first and named as MURR1 [12]. Later, it was renamed as Copper-metabolism Murr1 Domain (COMMD1) after its first known function in the regulation of copper homeostasis [35]. COMMD proteins are approximately 20 kDa proteins that all consist of C-terminal COMM domain and N-terminal HN domain [35,36]. COMMD6 is an exception: it comprises a typical COMM domain but only a very short HN domain. The structures of both the domains have been solved using X-ray crystallography for COMMD9: N-terminal domain folds into an α-helical sphere, whereas the C-terminal COMM domain forms elongated structure that in crystal dimerizes in an intertwined left-handed handshake orientation [36]. In solution, COMMD proteins form elongated homo- and heterodimers [36,37].

Based on the known structures of COMMD9, the novel structure prediction tool AlphaFold [38] can predict structures for rest of the commander complexes with high reliability score. The predicted structures are presented in Figure 1, as well as a predicted dimeric structure of COMMD9 constructed in PyMOL based on the dimeric crystal structure of its COMMD domain (PDB-code: 6BP6) and the full-length structure predicted by AlphaFold.

COMMD proteins together with CCDC22 and CCDC93 can be referred to as a CCC-complex [39]. CCDC22 and CCDC93 belong to the group of microtubule-associated proteins and they comprise an N-terminal calponin homology-like (NN-CH) domain and C-terminal coiled-coil domain [5,40]. Experimental structural information about CCDC22 and CCDC93 is very limited, and AlphaFold fails to present a credible structure for them (Figure 1). The secondary structure, though, has consistently a globular NN-CH-domain containing N-terminus and long α-helixes in unpredictable arrangement (Figure 1). The CCC-complex can be co-eluted as a 600 kDa complex in size-exclusion chromatography, when selected COMMD proteins, CCDC22, or CCDC93 are used as baits [4,41]. Using cross-linking mass spectrometry, COMMD9 has been shown to interact with the NN-CH domain of CCDC93 proteins [36]. 

VPS26C (also known as DSCR3 or DCRA), VPS29, and VPS35L (also known as C16orf62) form a retriever complex [10]. The retriever complex is expected to be a structural homolog of the retromer cargo-recognition complex and is likely to anchor the commander complex to cell membrane [3,10,41]. A Cryo-EM structure of retromer complex was recently solved in 5.7 Å resolution [42]. In the retromer, VPS35, the biggest protein in the complex, forms an alpha solenoid, a curved α-helical ribbon that binds VPS26A/B and VPS29 to each of its N- and C-terminals, respectively. This trimer can dimerize in both head-to-head and tail-to-tail arrangements and further form homopolymers. The sequence similarity between VPS35 and VPS35L is 11%, and VPS35L is predicted to fold to an alpha solenoid structure similar to VPS35. VPS26C shares a 17% sequence identity with VPS26A and 16% with VPS26B and is predicted to fold to two curved β-sheet sandwiches similarly to VPS26A/B. Both retromer and retriever mediate intracellular cargo, but they interact with different cargo adaptor proteins, SNX27 and SNX17, respectively [10,43]. The structures of VPS26C and VPS35L presented in Figure 1 are predicted by AlphaFold based on their homologs VPS26A/B and VPS35, respectively. As VPS29 belongs to retromer complex as well, its structure has been solved and deposited into the protein data bank (RCSB PDB).

DENND10, also known as FAM45A, belongs to a family of guanine nucleotide exchange factors (GEFs) that activate Rab proteins to coordinate intracellular membrane trafficking [11,45]. DENND10 was initially classified as non-classical member of the DENN protein family based on predicted structural homology, as there were no known GEF function for DENND10. Later, DENND10 was found to regulate Rab27a/b [45]. DENND10 comprises a single DENN domain, and its structure predicted by AlphaFold is presented in Figure 1.

Taking together, the COMMD proteins are known to form homo- and heterodimers, and COMMD9 is located next to the N-terminus of CCDC93 in a CCC complex. The retriever subcomplex can be predicted to assembly around alpha solenoid structure of VPS35L, but how the CCC-complex, retriever and DENND10 proteins are assembled to a functional commander complex remains an open question.

It is not clear whether COMMD proteins can act alone separately from the commander complex, and if CCC-complex is functional without the retriever subcomplex or is it always assembled as a full commander complex. COMMD genes are highly conserved in Metazoa [35]. Vertebrates possess all 10 COMMD genes, and they are 90% conserved in Mammalia. Individual COMMD genes are found in genomes of lower metazoans like insects and worms.

According to the Genome Aggregation Database gnomAD [46], all the commander complex proteins except CCDC22 and VPS29 appear to have a high resilience against point mutations. CCDC22 appears to not to tolerate mutations as it has no loss-of function variants in the database (pLI = 0). This indicates that normally functioning CCDC22 is essential for functional cell. VPS29 seems crucial as well with pLI of 0.75.

According to the Human Protein Atlas database, commander proteins have very low tissue specificity, and they are expressed ubiquitously in the human body (Figure 2a) [47]. With approved reliability scores (Protein atlas score based on data from RNA-seq, protein/gene characterization and immunohistochemistry) the expression of COMMD proteins 1, 3, 8, and 10, and CCDC93 is detected in almost all human tissues. The data on intracellular localization of the commander complex proteins is more extensive in the Human Protein Atlas database (Figure 2b). Proteins are mainly localized in cytosol and nucleoplasm, but they are found in plasma membrane and vesicles as well. As the proteins of commander complex can be found throughout the cell all over different tissues, it is not surprising that the commander complex or its individual proteins are involved in a wide range of fundamental cellular functions.

## 3. Endosomal Trafficking

As mentioned, the Commander complex plays a major role in cellular homeostasis maintenance. It participates in endosomal trafficking of transporter proteins between plasma membrane, trans-Golgi network, nucleus, and lysosomes along cargo networks. In addition, individual commander proteins can regulate channel activity directly or indirectly by enhancing or suppressing channel ubiquitination and thus their degradation.

The assembly of the commander cargo complex is illustrated in Figure 3. The cargo complex is initiated by a membrane-bound cargo adaptor sortin nexin 17 (SNX17) which mediates α_5_β_1_ integrin dependent trafficking [10]. SNX17 interacts with the commander complex via the retriever subcomplex, VPS26C being crucial for this interaction [10]. In endosomal cargo, commander complex works hand in hand with Wiskott-Aldrich syndrome protein and SCAR homologue (WASH) complex [3,11,48]. WASH complex subunit 2A, also known as FAM21, recruits the commander complex to the endosomes by binding VPS35L, the largest protein in retriever complex, and the C-terminals of CCDC22 and CCDC93 [10,39,43]. COMMD1, CCDC22, CCD93, VPS29 and VPS35 were shown to co-localize to early endosomes with the WASH complex in fluorescence microscopy experiments [39]. The retriever complex and the CCC-complex can possibly act separately in cargo, as FAM21 can interact individually with both retriever and the CCC complexes [10].

To recognize the signal molecules and to regulate trafficking, cargo adaptors need to recruit accessory proteins like the small GTPase Rab proteins [49,50]. A Rab5 homolog Rab7 binds retromer complex binding it to the membrane in early endosomes and Rab5 can regulate the membrane binding of retromer too [49,51]. Thus, it could be expected to act via retriever in the commander complex. However, it has been shown that COMMD5 and COMMD1 regulate endosomal trafficking directly: their COMM domains can bind endosome-bound Rab5 whereas N-terminal domains bind actin and tubulin [50].

DENND10 regulates endosome dynamics from early to late endosomes [45]. It binds several Rab proteins, and regulates endosome positioning via Rab27a and Rab27b, which in turn recruit downstream effectors for vesicle budding, organelle motility or docking at target compartments.

The intracellular cargo pathways are regulated by WASH-commander complex: WASH complex activates actin-related protein 2/3 (ARP2/3) complex which induces F-actin branching in cytoskeleton to build network for endosomal traffic (Figure 3a) [43,52,53]. Commander complex, when bound to WASH complex, inhibits the branching of F-actin [11]. COMMD5 can interact directly with actin and tubulin via its COMM domain and has been proven crucial for cytoskeletal F-actin stability [50]. Further on, CCC can activate Myotubularin Related Protein 2 (MTMR2) to regulate vesicle trafficking. MTMR2 converts phosphatidylinositol-3-phosphate PI(3)P to PI(4,5)P, of which (PI(3)P) interacts with FAM21 and PI(4,5)P further anchors COMMD1 to plasma membrane [16].

### 3.1. Ion Channels and Transporters

COMMD proteins are named after the first known function of COMMD1 in copper homeostasis regulation [35]: copper toxicosis in Bedlington terriers was linked to the absence of functional COMMD1 [54]. In a human, mutations in copper-transporting P-type ATPase ATP7A and ATP7B lead Copper deficiency disorder Menke’s disease and Wilson’s disease, where copper accumulates into body causing symptoms in liver and brain, respectively [28,55]. A mutation in COMMD1 protein leading to a loss of COMMD1-CCDC22 interaction or CCDC93 deficiency have both similar copper accumulating effect in cells [18,39,56]. In a healthy cell, COMMD1 interacts with ATP7A and ATP7B guiding their traffic to the plasma membrane to reduce intracellular copper levels [39,57,58]. COMMD1 binds copper, but it is not known if it is copper that triggers the transport of ATP7A and ATP7B to the plasma membrane [37].

COMMD1 participates in ion channel regulation in the kidney: it has been shown to regulate the endosomal trafficking and ubiquitination of amiloride-sensitive epithelial Na+ channel (ENaC) [13,59], cAMP-regulated CL- channel cystic fibrosis transmembrane conductance regulator (CFTR) [13], and Na-K-2Cl cotransporter (NKCC1) [15].

COMMD1, COMMD3, COMMD9 and COMMD10 inhibit ENaC reducing Na+ currents in the kidney epithelium [60,61]. COMMD1 binds ENaC via its COMM-domain, dragging the receptor from the cell surface to the endosomes [59]. COMMD1 and COMMD10 have been shown to suppress ENaC indirectly via Nedd4–2-dependent ubiquitination [61,62], most likely via SGK1 and Akt1 kinases [63]. This leads to the receptor cargo to peroxisomes for degradation. The regulation of a Cl- channel cystic fibrosis transmembrane conductance regulator (CFTR) by COMMD1 follows different pathway: an overexpression of COMMD1 reduces ubiquitination of CFTR and its recycling to the endosomes [14]. COMMD1 prevents the ubiquitination by directly binding to an intracellular loop of CFTR. COMMD1 can also bind directly to NKCC1 regulating its ubiquitination.

In addition to these multiple levels of ion channel regulation of COMMD proteins, there is evidence that intracellular ion levels can regulate COMMD protein expression. In kidney epithelial cells, COMMD5 expression level depends on calcium concentration [64]. This indicates that COMMD proteins may be profound regulators of renal homeostasis and more comprehensively blood pressure.

### 3.2. Cholesterol Intake

The commander complex, together with WASH, regulates low-density lipoprotein receptor (LDLR) traffic [17,18]. The COMMD domain of COMMD1 can interact directly with LDLR, which will then be transported to the cell surface by commander-WASH complex. COMMD1, COMMD3 and COMMD6 interact with WASH recruiting SNX17 to transfer LDLR to plasma membrane to intake cholesterol from plasma to cytosol. Impaired CCDC22 or COMMD1 were shown to cause hypercholesterolaemia in mammals, as in the ambiguity of LDLR on the cell surface cholesterol intake to the liver was prevented in humans, mice, and dogs [17,18]. It recognizes β-integrin and low-density lipoprotein receptor (LDLR), FAM21-CCC complex interaction being critical for this cargo [10,17,18]. CCDC22 also participates in cholesterol biosynthesis regulation.

### 3.3. Viral-Host Interactions

Recently, the commander complex was found to participate in the endosomal viral life cycle in a genome-scale CRISPR loss-of-function screen [30]. RAB7A, which anchors the commander to the cell membrane, is involved in the initial virus endocytosis. Commander proteins COMMD2, COMMD3, COMMD4 and CCDC22 were linked to endosomal recycling of viral particles. Earlier, the retromer complex has been shown to interact directly with viral envelopes during infection of human papillomavirus, HIV, and Chlamydia trachomatis [65,66,67]. Although VPS29 is known to play a role in the pathogen recognition [68], it is not known if these observations are specific for retromer complex or if retriever, and further, the commander complex would be involved.

## 4. Immune Response Regulation

COMMD proteins regulate NF-kB signaling pathways controlling immune reaction [19,21,35]. This has been reviewed widely before, so we discuss it here only briefly. COMMD8 bound to CCDC22, and COMMD10 have been shown to be able to bind cullin-RING-ligase (CRL) 1 to ubiquitinate I-kB and thus activate NF-kB. On the contrary, COMMD1 with CCDC22 bound to CRL2 inhibit I-kB ubiquitination which further leads to ubiquitination and down-regulation of NF-kB [19,20,21]. COMMD10 acts as immunosuppressor inhibiting NF-kB along COMMD1 [9,22], whereas COMMD7 may both inhibit NF-kB signaling [69] as well as upregulate it in complex with COMMD1 and NEMO [69,70]. The regulation of this feedback system remains unknown and it is not clear what roles all the individual COMMD proteins have in the system as a systematic research is missing.

There are bits and pieces of knowledge about other pathways where COMMD proteins regulate immune response. COMMD3 and COMMD8 are crucial for B-cell migration [23]. They recruit GPCR-kinases to activate β-arrestin mediated immune response in B-cells. COMMD5 reduces the upregulation of proinflammatory cytokines and chemokines, such as TNF-α, IL-1β, and monocyte chemotactic protein (MCP)-1 in mice kidney after trauma acting as immunosuppressor [31]. COMMD7 induces CXCL expression to inflict immune response [71]. As systematic research on the effect of COMMD proteins on the immune system is missing, it can be assumed that the immune response regulation by COMMD proteins expands beyond the current knowledge.

## 5. Cell Cycle and Cancer

COMMD proteins get localized into nuclei as well where they interact with several transcription factors participating in transcription and cell cycle regulation. COMMD1, COMMD5, and COMMD10 are known to act as tumor suppressors, and their expression levels are reduced in several cancers including cancers, seminoma, melanoma, as well as kidney, pancreatic, colorectal, and ovarian cancers [25,27,72,73,74]. Downregulation of COMMD1 leads to cancer malignancy [25]. On the contrary, COMMD9 acts as a carcinogen in lung cancer [27]. COMMD7 inhibits tumor growth in liver and kidney cancers [71,75] but can act as a carcinogen in hepatocellular carcinoma [68]. The overexpression of COMMD6 can either improve or worsen the cancer prognosis depending on the cancer type [33]. Survival from hepatocellular carcinoma is improved with high expression of COMMD1/4/10 and on the other hand, overexpression of COMMD2/3/5/7/8/9 decreased the survival [31].

COMMD1 acts as a tumor suppressor by inhibiting NF-kB and Hypoxia-inducible factor (HIF) mediated gene expression. An active HIF is a heterodimer of HIF-1a and HIF-1b. The COMM-domain of COMMD1 binds HIF-1a inhibiting its dimerization and further transcription activation of oncogenes [25,76]. COMMD10 binds p65 to inhibit NF-kB reducing the nuclear translocation of NF-kB [74]. Actin-associated protein FAM107A, also known as DRR1 has been shown to link COMMD1 to nuclear F-actin in neuroblastoma cells. This enhances COMMD1 mediated NF-kB degradation inhibiting G1/S phase transition [77]. COMMD10 as well inhibits NF-kB acting as a tumor suppressor, as shown with colorectal cancer [74].

COMMD1 regulates copper-dependent Cu,Zn superoxidase dismutase (SOD1) activity: in the presence of excess copper, COMMD1 binds SOD1 obstructing its homodimerization. This inhibits SOD1 activity which leads to increased levels of intracellular superoxide radicals [78]. COMMD1 may affect redox balance also via Guanine-rich RNA Sequence Binding Factor 1 (GRSF1) interaction [79]. COMMD7 is also involved in creation of reactive oxygen species in cytosol [80]. 

COMMD5 was previously known as the hypertension-related, calcium-regulated Gene (HCaRG) as its first known function was to regulate the growth of renal epithelia [24]. COMMD5 is a tumor suppressor as it promotes cell proliferation and kidney regeneration in mice after trauma [81]. This happens via p53 independent p21 activation, where inhibition of G1/S phase transition prevents cell growth [31,32]. Tumorigenesis of COMMD5 deficiency is ascribed by cell growth, but also uncontrolled cell invasion as COMMD5 is required for cell migration regulation [50]. COMMD5 as a tumor suppressor regulates several tumorigenic kinases: it inhibits epidermal growth factor receptor (EGFR) expression and phosphorylation in kidney cancers and suppresses the MAPK and PI3K/AKT signaling pathways in renal carcinoma and melanoma cells [32]. Furthermore, overexpressed COMMD5 drags EGFR to endosomes reducing its activity in consistent with DENND10 [50]. A similar function has been reported for DENND10 [45].

COMMD9, unlike other COMMD proteins, was found to be overexpressed in several lung cancer cell lines [27]. The oncogenic function of COMMD9 is mediated via activation of transcription factor DP-1 (TFDP-1) which promotes oncogenic E2F1 and p53 transcription factors to transit cell cycle to S-phase [27]. Knocking down COMMD9 trapped cells to G1/S transition preventing cancer growth. COMMD7 acts as an oncogene increasing metastatic properties of hepatocellular carcinoma [75]. It activates interferon gamma-induced protein 10 (IP-10) that promotes cell growth via inhibition of anti-apoptotic Cyclin D and Bcl-2 as well as enhancement of pro-apoptotic Bax in adenocarcinoma cells.

We report here several individual findings on the function of the COMMD proteins in different types of cancer. Some of the data comes from research using cancer cell lines, and some from cancer patient data. Many of the COMMD proteins seem to play a role both as oncogenes and tumor suppressors, and the factors regulating this dual function remain unknown. This data indicates, that COMMD proteins are important players in cancer biochemistry, and most likely a major part of their functions in cell cycle regulation remains to be found.

## 6. Conclusions

Commander complex is a recently discovered protein complex that participates in endosomal trafficking regulating intracellular homeostasis and cellular signaling in most human tissues. Based on the current understanding, the individual proteins of the commander complex act alone or together as a complex regulating gene expression to control cellular functions such as cell growth, cell cycle, cell death, various signal transduction pathways, and immune response. These activities are all very fundamental for normal cell functioning and are thus linked to a variety of different diseases. This review summarizes the known effects of commander proteins on cell signaling. It becomes evident that the commander complex is a more important player than has been understood, and there is demand for an extensive, systematical investigation of the cellular functions of the commander complex.

## Figures and Tables

**Figure 1 cells-10-03447-f001:**
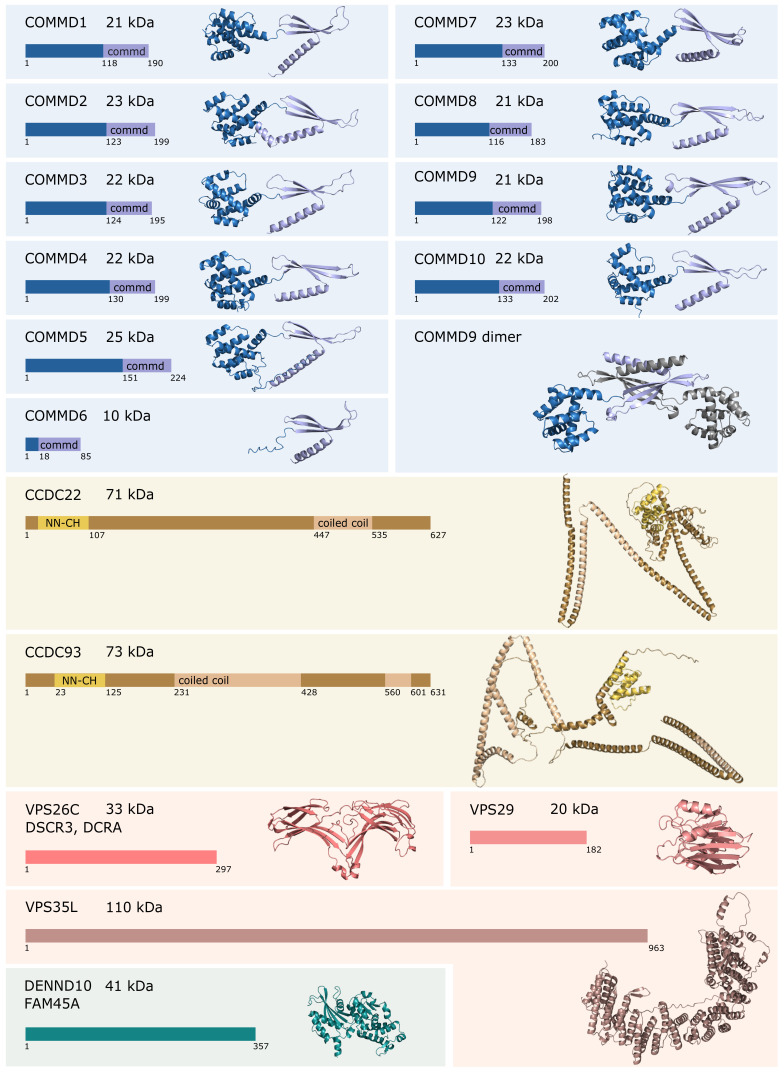
The predicted structures of commander complex proteins. The protein structures calculated by AlphaFold are presented in cartoon representation. Specific domains have been color coded according to the schematic presentation of the protein sequence on the left. VPS29 structure is from experimental data (PDB-code: 5GTU). The dimeric structure of COMMD9 has been generated combining the AlphaFold structure of COMMD9 with the crystal structure of COMMD9 COMMD domain (PDB-code: 6BP6) in PyMOL [44]. The second chain is shown in gray for clarity.

**Figure 2 cells-10-03447-f002:**
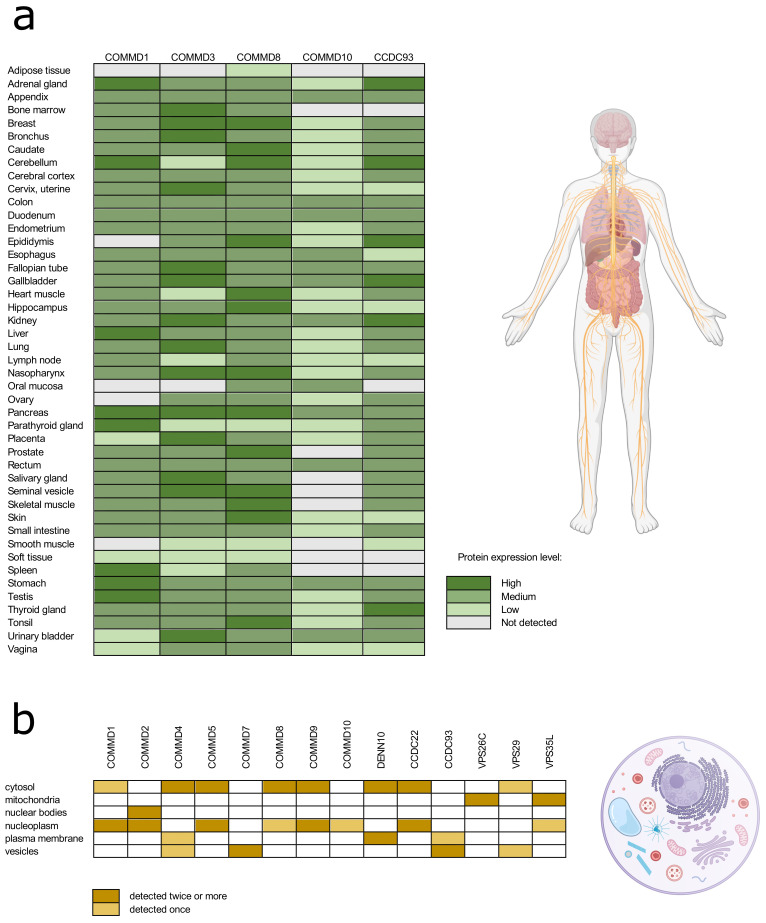
The protein expression in different tissue types (**a**) and intracellular location (**b**) of the commander complex proteins according to the Human Tissue Atlas database. Data with reliability score less than approved were excluded from the table.

**Figure 3 cells-10-03447-f003:**
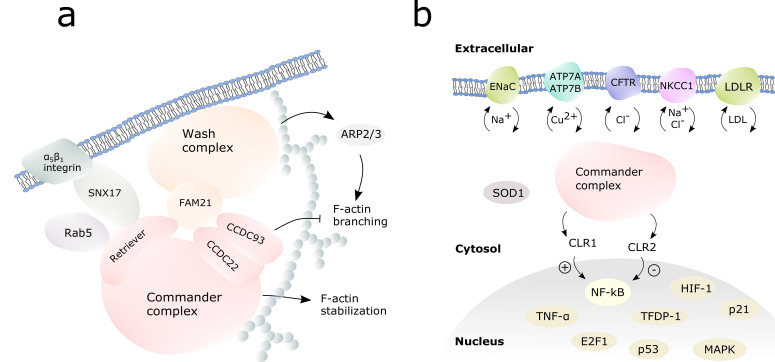
Commander complex in cargo and cell signaling. (**a**) Assembly of the commander-WASH complex for the endosomal cargo. The WASH complex is shown in orange, the commander complex in pink. White spheres represent F-actin. (**b**) The commander complex (pink) regulates the cargo of transmembrane channels to the cell surface and back, as well as the activity of selected transcription factors in the nucleus.

## Data Availability

Not applicable.

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
