# Peer review of "Commander Complex—A Multifaceted Operator in Intracellular Signaling and Cargo"

_cells, 2021, doi:10.3390/cells10123447_

Round 1
Reviewer 1 Report
The authors describe the commander complex and its relatively poorly known functions in inflammation and cancer. Over all, a very nicely review and overview of the various functions of the commander complex.
Minor remarks:
There is a problem with reference formatting [Error! Reference source not found]. Please fix those.
Author Response
Dear reviewer,
thank you for your positive feedback.
We have performed a spell check for the manuscript and edited the reference list to find any errors in the formatting.
Reviewer 2 Report
Laulumaa and Varjosalo present here a nice review on the « Commander complex - a multifaceted operator in intracellular signaling and cargo ».
The authors first present the different proteins of the complex and how they assemble. They also use the protein atlas data for tissue and cellular localization of the different proteins forming this complex. Three major functions of these proteins are depicted : endosomal trafficking, immune response regulation and cell cycle/cancer.
Previous data is depicted and illustrated but the review would gain with the addition of some elements to the ones presented here.
Figure 1 illustrates the predicted structures of the different components of the complex. The figure can be rendered more informative:
- Addition of the full official gene names and other names for a better linking with previous data is required. This could be mentioned on Fig1. DENN10 is also named DENND10 or FAM45A for example.
- An illustration of the complex could be useful to understand how the different partners interact with each other.
- The protein AA length can be mentioned
- Known domains of some proteins are mentioned. This could be done for all the proteins when characterized.
Some paragraphs require further clarification.
- L92-L102: Can the differences between VPS35L and VPS35 be further explained. What distinguishes VPS26C from VPS26A/B? Are these redundant proteins?
- Why is DENND10 a “non-classical” GEF?
- the Retriever complex being key in the field, can an illustration be added to better understand this part? This would also help understanding the Fig3a as the proteins that form the retriever are not mentioned.
- the authors state that “CCDC22 and VPS29 have a high resilience against point mutations. CCDC22 has no known loss-of-function variants.” Can they further develop on this aspect as the RTSC2 syndrome has CCDC22 mutations.
The authors use Protein atlas database for Figure 2. Is the data presented here further corroborated with recent PubMed publications? The sub-cellular localizations should be further documented with recent research (as in Mallam and Marcotte, 2017).
In the Viral-host interactions part, the authors state Chlamydia trachomatis. This is confusing as this bacteria is associated to “viral envelopes”. Are there bacteria-host interactions that involve some COMMD proteins too?
The last paragraph requires clarification as there are contradictory statements. In liver, what is the role of COMMD7? Is it a tumor suppressor or a carcinogen?
Minor points:
. Many referencing problems : « References source not found »
. Spelling
L72: “dimerdic” for dimeric
L103: member “of” rather than “to”
L107: remains “an” open question
L111: “Vertebrate” requires an “s”
L279: “reactive” oxygen species
